# The detection of algebraic auditory structures emerges with self-supervised learning

**Pierre Orhan** [1*], **Yves Boubenec**[1*], **Jean-Rémi King**[1,2*]

**1** Laboratoire des Systèmes Perceptifs, Département d'études Cognitives, École Normale Supérieure, PSL University, CNRS, Paris, France, **2** Meta, Paris, France

\* pierre.orhan@ens.psl.eu (PO); jeanremi@meta.com (JRK); yves.boubenec@ens.fr (YB)

**Data availability statement:** Data necessary to reproduce this study are the model weights and exact pretraining datasets, which can be found

## Abstract

Humans can spontaneously detect complex algebraic structures. Historically, two opposing views explain this ability, at the root of language and music acquisition. Some argue for the existence of an innate and specific mechanism. Others argue that this ability emerges from experience: i.e. when generic learning principles continuously process sensory inputs. These two views, however, remain difficult to test experimentally. Here, we use deep learning models to evaluate the factors that lead to the spontaneous detection of algebraic structures in the auditory modality. Specifically, we use self-supervised learning to train multiple deep-learning models with a variable amount of either natural (environmental sounds) and/or cultural sounds (speech or music) to evaluate the impact of these stimuli. We then expose these models to the experimental paradigms classically used to evaluate the processing of algebraic structures. Like humans, these models spontaneously detect repeated sequences, probabilistic chunks, and complex algebraic structures. Also like humans, this ability diminishes with structure complexity. Importantly, this ability can emerge from experience alone: the more the models are exposed to natural sounds, the more they spontaneously detect increasingly complex structures. Finally, this ability does not emerge in models pretrained only on speech, and emerges more rapidly in models pretrained with music than environmental sounds. Overall, our study provides an operational framework to clarify sufficient built-in and acquired principles that model human's advanced capacity to detect algebraic structures in sounds.

## Author summary

Experimentalists have repeatedly observed a human advantage in the detection of algebraic structures, notably through auditory paradigms. This ability to detect structure is thought to be key to the emergence of complex cognitive operations. Yet, it remains debated if this ability is discovered or innate in the form of a specific mechanism. In this

at: https://huggingface.co/NDEM. We share our code for protocol generation: https://github.com/PierreOrhan/ControlledStim. As well as for the probing of the model surprise: Repository for model probing: https://github.com/PierreOrhan/SurpriseProbing.

**Funding:** This project has received funding from the European Union's Horizon 2020 research and innovation program under the Marie Sklodowska-Curie grant agreement No 945304 to PO. This work was funded in part by FrontCog grant ANR-17-EURE-0017 and ANR-10-IDEX-0001-02. The funders had no role in study design, data collection and analysis, decision to publish, or preparation of the manuscript.

**Competing interests:** The authors have declared that no competing interests exist.

article, we show how a model progressively and spontaneously learns to detect auditory structures. The model replicates several experimental findings but only under certain developmental conditions. Notably, exposition to music or environmental sounds, but not speech, is sufficient for the emergence of algebraic structure detection. As a result, this work proposes self-supervised learning as a developmental model of abstract cognitive abilities.

## Introduction

**Theorizing humans' exceptional ability to detect structures.** Humans have a unique ability to detect abstract structures from sensory input: Unlike other animals, children spontaneously detect syntactic trees underlying speech input [1–5], and this ability seems to go well beyond language [3,4,6,7]. For example, we spontaneously detect the temporal structures underlying tone sequences [8], the geometrical structures underlying drawings [9,10] and the hierarchical relationships between semantic concepts [11]. Such structure detection is at the heart of a tension between two competing views: "Rationalists" argue that an innate and specific mechanism, like "Merge" [12] or "a recursive neural code" [10], allows the human brain to instinctively represent information as tree structures. On the other hand, "Empiricists" argue that this cognitive ability does not require a specific, nor an innate mechanism, but rather emerges from an efficient statistical learning of naturalistic inputs [13].

**Brain and behavioral bases of minimal structure building.** To confront these perspectives, a variety of minimalist protocols focusing on the brain and behavior's fundamental aspects of structure building have been proposed [14–19]. Focusing here on minimalistic auditory structures devoid of linguistic features, several protocols have been designed to elucidate whether, when and how abstract temporal structures are represented in the developing mind of children. These protocols typically use artificial stimuli to ensure that participants are not already familiar with the tested structures. In particular, Saffran et al; [14] observed that children as young as eight-month-old spontaneously segment recurring sequences of syllables. In adults, Al Roumi et al. demonstrated that the ability to detect auditory structures strongly depends on their algebraic complexity [8]. These spontaneous behavioral abilities do not appear to be associated with a unique neural system: First, the violation of simple auditory regularities elicit an early mismatch negativity (MMN) [20] in electro-encephalography (EEG) recordings, while more complex auditory structures are associated with a late positivity (P300b) [21,22]. Second, the detection of long auditory sequences depends on a sustained, distributed, and elevated Magneto-Encephalography (MEG) response [23]. Finally, the complexity of auditory structures is linked to parietal activations, and seems dissociated from the language network in the brain [8].

**Modeling structure building.** Consequently, a variety of neural or symbolic models have been proposed to account for these behavioral and neural observations. Local plasticity in the thalamo-cortical circuit has been proposed to account for MMN in auditory irregularities [24]. Prediction by Partial Matching (PPM) models, like Information Dynamics Of Music (IDyOM) [25] model or PPM-decay model [26], have been shown to provide optimal predictions for melodic expectation, and to mirror human behavior in several tasks [27]. In addition, a variety of neural network models have been proposed to tackle systematic compositions of symbolic input sequences [28–32]. The common – although not always explicit – denominators across these efforts, is the idea that discovering the specific mechanism responsible for spontaneously detecting structures would prove critical to build systems that act and thinks like humans. Connectionists [33] have indeed advocated for learning mechanisms

and architecture that are not based on sets of explicitly coded statistical models of structures. Instead, the architecture, the learning algorithm, and the environment should constrain the computations and structures emerging in the model.

**Main challenges.** Overall, however, these efforts face a common challenge: all models are crafted to operate exclusively on one experimental protocol. This approach thus presents two major limitations. First, no model has been systematically tested on its ability to account for a large variety of experimental observations. Second, no model leverages the possibility of learning structures from natural stimuli. Detecting artificial structures with such a restricted approach thus constrains us to either design a hard-wired system or to extensively train it with our artificial stimuli. In sum, the rationalist view benefits from an intrinsic head-start.

**Modern AI to the rescue of empiricists.** Here, we argue for an alternative approach. If structure building emerges from learning, then we should investigate models that can learn from naturalistic stimuli, and then only test their ability to detect artificial structures – not the other way around. Following the recent observation that large language models (LLMs) demonstrate remarkable generalization abilities without additional training [34–36], we hypothesize that the representations modern AI systems can learn from naturalistic data may suffice to lead them to spontaneously build adequate structures when presented to artificial structures like algebraic sequences. To avoid LLMs, and their unreasonably large and symbolic datasets, we here focus on self-supervised learning (SSL) architectures designed to process natural sounds. Previous works, in vision [37,38] and audition [39], have shown that such architectures develop human-like representations from a similar data regime. Pretraining on natural datasets has been shown to promote representations that favor the emergence of abstract structure detectors. Notably, [40], demonstrated that letter detectors could be learned over features learned from natural images. Although these detectors were robust across letter fonts and noise levels, they emerged after exposure to letters. In this work, we investigate the emergence of a structure detection algorithm purely after exposure to natural sounds. Moreover, we investigate the factors and dynamics of the emergence of such detectors.

**Approach.** Specifically, we use a standard SSL architecture, Wav2vec2 [41], to train a series of deep neural networks to unmask natural sounds. We then present these models to four psycho-acoustic protocols developed for structure detection, namely: [14]'s syllable chunking paradigm, [23]'s RangReg paradigm, [21]'s Local-Global and [8]'s Algebraic Patterns paradigm. To compare these models to humans, we evaluate the models' ability to spontaneously detect auditory structures by measuring their surprise in response to each sound, and in particular when the auditory regularities violate the artificial structure. Critically, we systematically investigate how the type and the amount of pre-training impact the models' ability to spontaneously detect auditory structures. Overall, our results reveal the conditions in which an efficient SSL architecture exposed to natural sounds may suffice to account for humans' spontaneous ability to detect algebraic sequences. Without a generalization test, it is not possible to evaluate whether the model learns and uses algebraic structures in a rule-like manner. Consequently, we implement such a generalization test and demonstrate that the models effectively generalize to novel sounds.

## Approach

We aim to evaluate whether a model trained with a reasonable amount of naturalistic sound stimuli, learns latent representations that allow it to spontaneously detect the structures of variably-complex algebraic or probabilistic sequences.

We focus on the auditory modality, where the detection of algebraic structures has been more systematically investigated than the visual modality.

**Model.** We study a self-supervised neural network: Wav2vec2 [41]. The model takes as input sound waveform. The model is structured hierarchically with local filtering (convolutions) followed by global contextual processing (transformer) (Fig 1B). The model's objective is to predict masked parts of the sounds. More precisely, acoustic latent computed by the convolutions layer are first randomly masked. Based on the surrounding context, the model then tries to change this mask latent vector back to the original vector. The model loss contrasts this reconstruction with the target and a set of other latents used as negatives.

**Pretraining.** The model is exposed (pretraining) before the experiments to several epochs of a modest (900 hours) sound dataset and optimized to diminish its contrastive loss by predicting masked latent acoustics for a maximal amount of 100,000 gradient steps. We first pretrain the model on a general audio dataset, composed of $3 \times 300$ hours of public datasets: Librispeech (speech, [42]), FMA (music, [43]) and Audioset (environmental sounds [44]). We removed from the Audioset dataset, musical and speech sounds thanks to the accompanying taxonomy. This partially removed all music and speech sounds, with errors (6% of music and 9% of speech) remaining due to the poor annotations of the Audioset data. To investigate

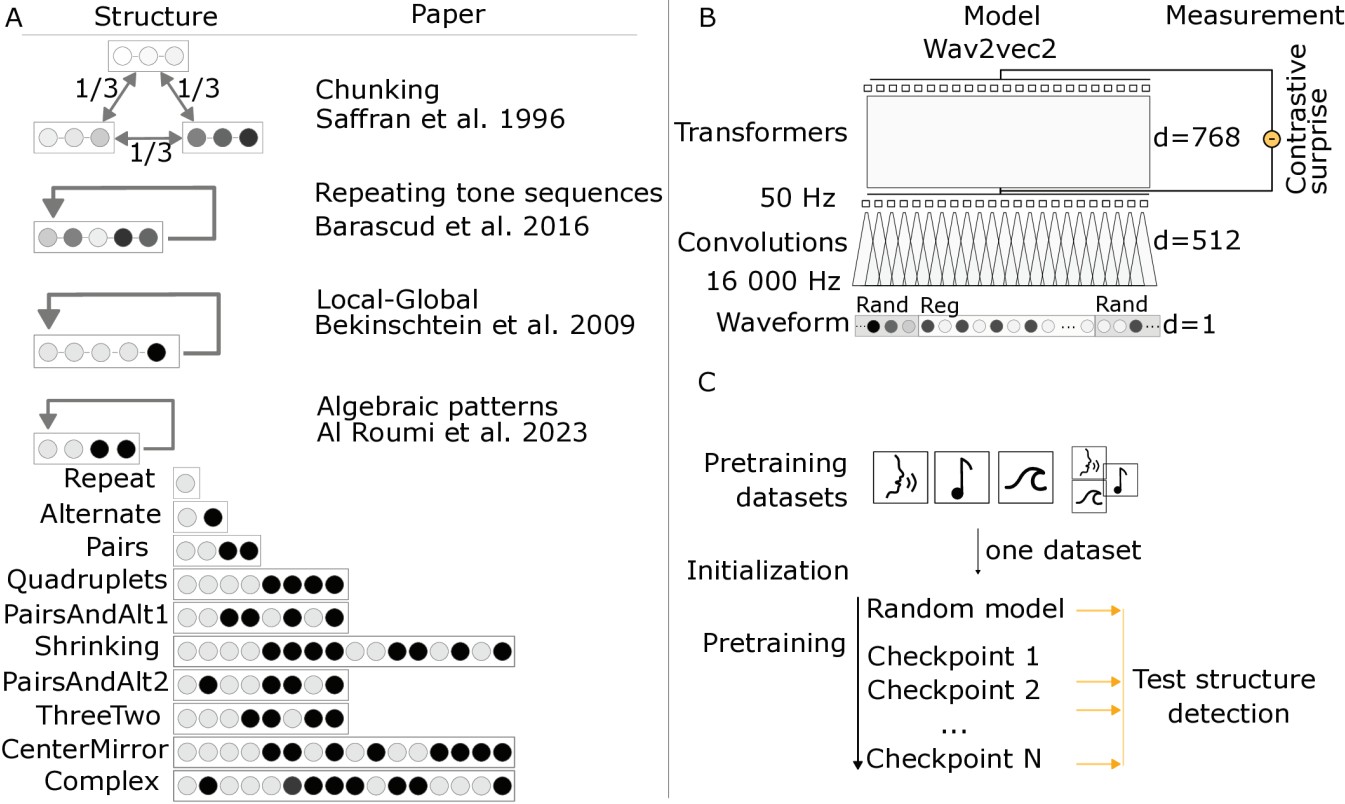

**Fig 1. A: schematic of the structure of the sounds tested over the experiments.** The right column indicate the experimental paper testing how humans detect the corresponding structure. B: schematic presentation of the Wav2vec2 model used. d refers to the dimensionality of the features at different layer. The model takes waveform as input so d = 1 at the beginning. We measure the model's surprise as the contrastive loss to each sound elements in the sound. This loss is measured by masking 20 ms (50 Hz) latent vectors whose receptive field overlap with the sound element. This operation is repeated to get the model surprise of each sound element. C: Models are initially pretrained on one dataset among four possible: one for speech - Librispeech [42], one for music - FMA [43], one for environmental sounds - Audioset [44] deprived of speech and music and one that combine a subset of those. To see if and when structure detection emerge, we test the models on many checkpoints throughout the pretraining. Modeled studies: [8,14,21,23].

the effect of each dataset, we then pretrain the model separately on the full individual public datasets (951 hours for Librispeech, 587.9 hours for FMA, 622.1 hours for Audioset). The model is optimized with the original, base, parameters.

## Experimental paradigms

**Protocols.** We explore four experimental protocols (Fig 1A) [8,14,21,23] investigating the behavioral and brain foundations of structure detection. These experiments assess subjects' perception of structured sound sequences, exposing them to regular, random, or deviant sequences. For simplicity, we standardize this variety of experimental protocols as follows. First, the pretrained model is exposed to random sounds. This allows us to evaluate its 'basal' surprise values. Second, we present it to a repeated structure (i.e. a regular sequence, generated from a subset of a larger alphabet), which allows us to verify the diminution of surprise. Finally, we present a random sequence to verify that the change in surprise is not just due to time (e.g. fatigue). Sounds of this ending random sequence are drawn from the same set as the regular sequence, controlling for the effect of a diminishing alphabet, and instantaneously violating the preceding regular structure. We generate each rand-regular-rand sequence as a single waveform and present the whole sequence to our models. Presenting a single waveform is necessary as it enables the model to evaluate the probability of each sound element given the preceding context.

**Evaluation.** To reveal if the model detects the regularity, we compute the model surprise on each sound element (tones or syllables) (Fig 1B). The surprise is measured as the contrastive loss of the model prediction in response to the masking of each sound element. Note that the model is consequently evaluated on the same audio file as many times as there are sound elements in the file, with the mask positioned each time over a different, probed, sound element.

The model is evaluated in a "zero-shot" scenario, where it is presented with sound sequences but can't optimize anymore its parameters.

**Structures taxonomy.** The four experiments presently considered focus on different types of auditory structures ([4] for review). First, [14] investigated auditory 'chunks', a repetition of a short acoustic motif (e.g. ABCDABCD). Second, [23] extend this to longer and more complex sequences. As argued by McClelland and Plaut (1999), this extension limits the possibility of detecting regularities solely from acoustic cues (e.g. D->A transition) and instead requires memorizing the sequence or its structure. Third, [21] introduced two levels of auditory regularities to investigate the detection of local and global irregularities. Finally, [8] extended this approach to auditory structures with a varying degree of complexity. For all of these auditory sequences, two strategies can be undertaken to learn the sequence: either one simply memorizes the auditory sequence (e.g. XXY), or it identifies its underlying structure (namely: two repetitions of the same sound followed by a distinct one). To distinguish these two strategies, one can test whether the model generalizes to new auditory sequences, identical in their structure, but different in their sounds (e.g. AAB, CCD, etc).

## Results

## Statistical chunking of words

We tested if pretrained models could spontaneously discover words in a syllable stream. We replicated in-silico the experiment of Saffran et al. [14], which demonstrated that 8-month-old children can rapidly (less than 2 minutes) learn to detect 3-syllable words in a stream of

syllables. A random stream of syllables was followed by a regular stream of syllables composed of 4 randomly alternating 3-syllable words (Fig 2A).

**Structure detection.** We measured the model surprise to each syllable of the stream without changing the model weights (i.e. frozen model). As expected, the model surprise was high on the first part of the stimuli, composed of randomly alternating syllables (Fig 2B). Critically, the models' surprise dropped when the stream became regular and groups of three syllables repeated to form four possible "words". Notably, the model loss dropped on the second and third syllables of each word (Fig 2B). The model loss remained high for the first syllables, indeed the first syllables should not be predictable as words are randomly alternating in the syllable stream. The model is not trained: its internal weights are frozen. However, the model does need to observe, at inference, a few repetitions of the word to correctly predict its second and third syllables.

**Learning dynamic.** We then investigated the learning dynamic of this ability. We repeated the experiments for 371 checkpoints spanning the model's pretraining. We measured the difference between the model surprise during the regular stream, containing syllables grouped by words, and the ending stream, containing random sequences of syllables (Fig 2C). Overall, the model surprise to the random stream increased quasi-monotonously with the amount of pretraining. Together, these results demonstrate that passive exposure to natural sounds led the model to spontaneously detect repeated chunks of syllables without any additional training or specific built-in mechanism.

**Controls.** To investigate further the experiments of Saffran et al. [14], we also tested the models' reaction to a deviant group of three syllables. The deviant was either a never-seen non-word (i.e a sequence of syllables that had never been seen) or a part-word i.e a non-word that had been seen (the last syllable of a word followed by the first two syllables of another word). Similar to the eight-month-old children tested in Saffran et al. [14], the model's surprise peaked significantly to these two types of deviants (Fig A in S1 Material). This confirms the model's ability to perform in-context discovery of words in this particular protocol, as humans.

## Detection of repeated tones sequences

Since the model can detect repeated 3-syllable motifs, we questioned if this ability was restricted to artificial syllable streams or could extend to longer and more complex sequences. We leveraged a paradigm based on rapid sequences of tones developed by Chait and colleagues. Using this paradigm, Barascud et al. [23] showed that humans were able to optimally detect transitions from random to repeated sequences of 5, 10 or 15 tones and sub-optimally for 20 tones.

To compare our model to humans in this "Repeating tones" task, we modified the syllable approach described above as follows. As in Barascud et al. [23], tones were randomly sampled from a set of 20 tones covering a logarithmic frequency range from 222 to 2,000 Hz. A random stream of tones was followed by a regular stream, which repeated a sequence of tones and ended by a new random stream, alternating tones composing the repeated sequences (Fig 2D). We tested repeated sequences of length 5 and 20.

**Structure detection.** The model was able to detect the transition from regular to random for all sequences irrespective of their length (Fig 2E): i.e. its surprise was greater on the first random deviant tones than on preceding tones.

**Learning dynamic.** This regularity detection emerged rapidly during pretraining (Fig 2F): it developed over the first "year" of sound exposure, with the detection of short sequences detection preceding that of long sequences.

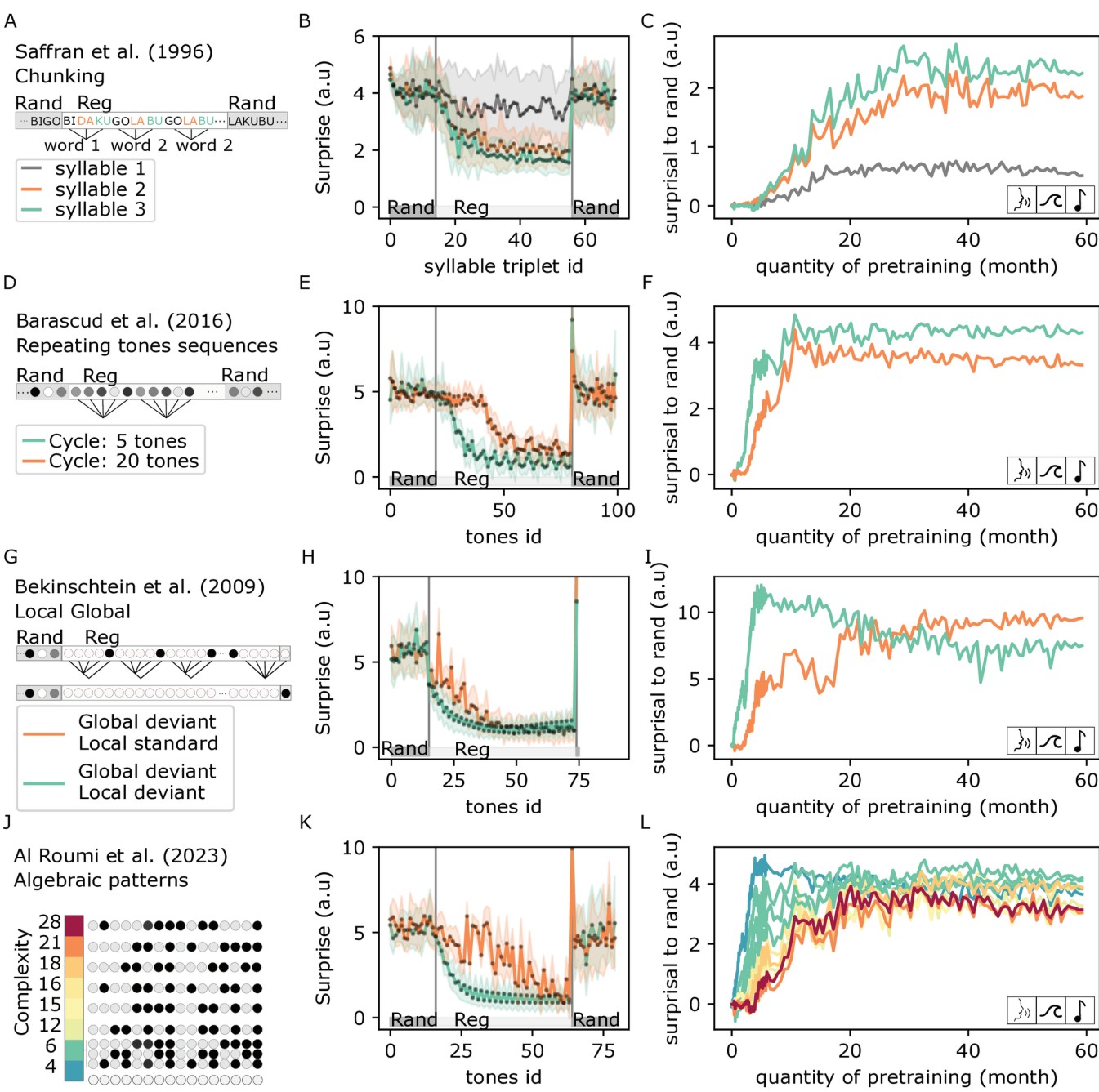

**Fig 2. A–C) Zero-shot emergence of word chunking. A.** Example of a syllable stream tested. **B.** Model contrastive loss to the first, second and third syllable is measured for each triplet of syllables (words) in a syllable stream. The loss is averaged over 30 trial of the tasks. The loss standard deviation across these trial is indicated as shaded area. (see Fig D in S1 Material for zooms). **C.** Evolution of the model ability to detect regular words as a function of its pretraining. This ability is measured by the difference between the mean contrastive loss of the second random sequence and the last repetition of the regular sequence. **D-E-F.** Same as A-B-C but with repeated tones sequences of cycle size 5 or 20 tones. **G-H-I.** Same as A-B-C but with a nested algebraic pattern from the Local Global paradigm. **J-K-L.** Same as A,B,C but with a set of 10 algebraic patterns of increasing complexity. In panel K, the "alternate" and "center-mirror" sequences are plotted, of complexity respective 6 and 21. Modeled studies: [8,14,21,23].

**Controls.** To detect repeated tones, the model could use the difference of acoustic between the repeating and the surrounding random sequences. Two arguments can be made against this hypothesis. First, detection happens on the first random tone and therefore is not using the long-term integration necessary to detect changes in acoustics. Second, we found that the model detected a single-tone deviation in the repeating sequence (sensitivity to deviant, D', of 2.5, see methods for computation of D', Fig G in S1 Material), indicating that the model is accurately tracking the exact sequence of tones, and not long-term acoustics.

As discussed in the original experimental paper [23], the transition from a random to a regular sequence can only be detected by a causal system after the second motif appearance. However, Wav2vec2 uses a bidirectional transformer and is therefore non-causal. The original Wav2vec2 [41] therefore detects the transition from RAND to REG at the first tone of the REG sequence (Fig B in S1 Material), which is, by definition, different from human. For all experiments in the paper, we made the model partially causal by preventing it from using future information (see methods). This change recovered a human-like regularity detection with the surprise diminishing during the second repetition only.

Based on this observation, we can state that the model is able to detect syllabic regularities, but also repetition of arbitrary tone sequences.

## Detection of global deviant in tone sequences

We then replicate the Local-Global paradigm which has been extensively studied both in humans [45–49] and animals [50,51]. In this paradigm, two 5-tone sequences are used: a XXXXX sequence (local standard) and a XXXXY sequence (local deviant). These sequences are presented repeatedly to the subject, forming a global regularity. A 5-tone sequence is considered a "global standard" if it is identical to the preceding 5-tone sequences. Structure detection is then tested by measuring the subjects' response to a global deviant sequence – i.e. a 5-tone sequence which is different from the preceding 5-tone sequences. If XXXXX is repeated multiple times, then the global deviant sequence is XXXXY, i.e a local deviant. Contrarily, if XXXXY is repeated multiple times, the global deviant sequence is XXXXX, i.e. a local standard. Therefore, this paradigm orthogonalizes local and global regularities. Experimentally, local deviants elicit rapid mismatch negativity (MMN), whereas global deviants evoke a late (300 ms) electro-encephalogram potential [21].

**Structure detection.** We evaluated the model surprise to tones of a rand-regular sequence, where the last tone is changed into its binary opposite (Fig 2G). The model detected the simpler global deviant - local deviant (Fig 2H). Remarkably, it also detected the global deviant - local standard (Fig 2H). Results are unchanged if we change the rand-regular-deviant to a rand-regular-rand sequence.

**Learning dynamic.** The global deviant - local deviant was detected after 3 weeks of pre-training, while it took around 3 months for the global deviant - local standard to be detected (Fig 2I, see Fig D in S1 Material for a zoom). We can't make a one-to-one temporal mapping between these models' dynamics and developing brains, as we present sounds without a break and in batch of inputs. Nevertheless, it is remarkable that children at 3 months old are also able to learn such global deviance [47]. Although the synthetic environment differs considerably from the continuous sound stream heard by the children, it remains that the model can learn from at least 3 months of data, which is reasonable at least in order of magnitude.

### Effect of sequence complexity on structure detection

Since global deviants are harder to detect than local deviants, we wanted to investigate precisely how a structure's complexity modulates its detection. Indeed the previous experiments do not distinguish if the model compresses or purely stores the sound sequences, with an added cost for longer sequences. If the model were to compress the sequence, the structure detection would be faster for sequences that are simpler to compress independently of their size. Remarkably, humans do compress rather than purely store algebraic tone sequences. Indeed, Al Roumi et al. [8] observed that in humans, binary sequences with simple structures were memorized more easily than sequences with complex structures. More precisely, they proposed a set of 10 binary sequences, each composed of 16 tones (see methods for details on the complexity metric). Remarkably, human detection performance was strongly correlated with the sequence complexity. We test the model on these algebraic patterns by embedding them in the rand-regular-rand protocol, as for repeating tones sequences.

**Structure detection.** The model detected the structure of sequences of diverse complexity (Fig 2K). Interestingly, the model surprise converged more slowly for the complex sequences: i.e. more tones are required for the model to change its surprise from the random baseline.

**Learning dynamic.** The model's ability to detect algebraic patterns emerged during pretraining (Fig 2L). Remarkably, this emergence appeared in sequential order, with sequences composed of chunks being accurately detected earlier (1–2 months, complexity ≤6) than sequences composed of nested structures (3–4 months, complexity ≥12) (see Fig D in S1 Material for a zoom).

**Controls.** The model's performances can be partially explained by the emergence of a trivial "copy" algorithm, where the model detects repetitions of an acoustic pattern instead of an actual algebraic structure. To verify that the model effectively represents sequences as algebraic structures, we tested its ability to generalize to novel sequences. In the regular part of these sequences, the 16 tones sequence is repeated 3 times with two novel distinct tones forming the sequence at each repetition. If the model detects the underlying structure, its surprise should progressively diminish with the number of repetitions. This novel experiment determines whether the model is a pure detector of acoustic repetitions or if it uses the algebraic structures of the sequence. In this "generalize" protocol, we also observed that the model could detect most structures and that this ability emerged through pretraining (Fig E in S1 Material). This demonstrates that the model detects abstract structures beyond the repetition of acoustic sequences. The two most complex structures were not generalized by the model.

To test whether our model learns the latent structure of auditory sequences, or simply memorizes successive tones, we perform a sequence-generalization test. Specifically, we evaluate it with auditory sequences that have the same structure but different sounds (e.g. XXXXYYZYYXXX) compared to those used in training (e.g. AAAABBCBBAAAA). In other words, each repetition of the auditory structure uses different tones. Our results show that our model still detects the violation of auditory regularities in these new sound sequences. This result demonstrates that the model does not simply memorize sound sequences, but detects their latent structures (Figs E and N in S1 Material).

### Music but not speech pretraining accelerates emergence of regularity detection

The above results show that pretraining is essential to the emergence of structure detection. In practice, the models were pretrained with a variety of sounds: speech, environmental sounds, and music. As these sounds have different statistical properties, it remains unclear which were used to support the emergence of structure detection. To address this question, we repeated

our analyses on three sets of models pretrained solely on speech, environmental sounds, and music, respectively. Two striking observations result from this analysis. First, exposition to speech alone did not allow the emergence of "zero-shot" structure detection (Fig 3A). Second, music exposure accelerated the emergence of structure discovery compared to environmental sounds (Fig 3B and 3C).

To confirm this difference between speech and music models was not explained by a difference in test stimulus, we extend our tests to a 3x3x4 design. In this design, all 3 models (music, speech, environmental) are tested with all 3 types of sounds (novel excerpts of environmental sounds, artificial syllables, and tones) on all 4 protocols. To summarize our results, we report in Table A in S1 Material the surprisal to rand metric (similar to Fig 2C, 2F, 2I, 2L) for each of these conditions. Speech models systematically failed to perform structure detection in all 3x4 conditions, in stark contrast with the music and environmental sounds model.

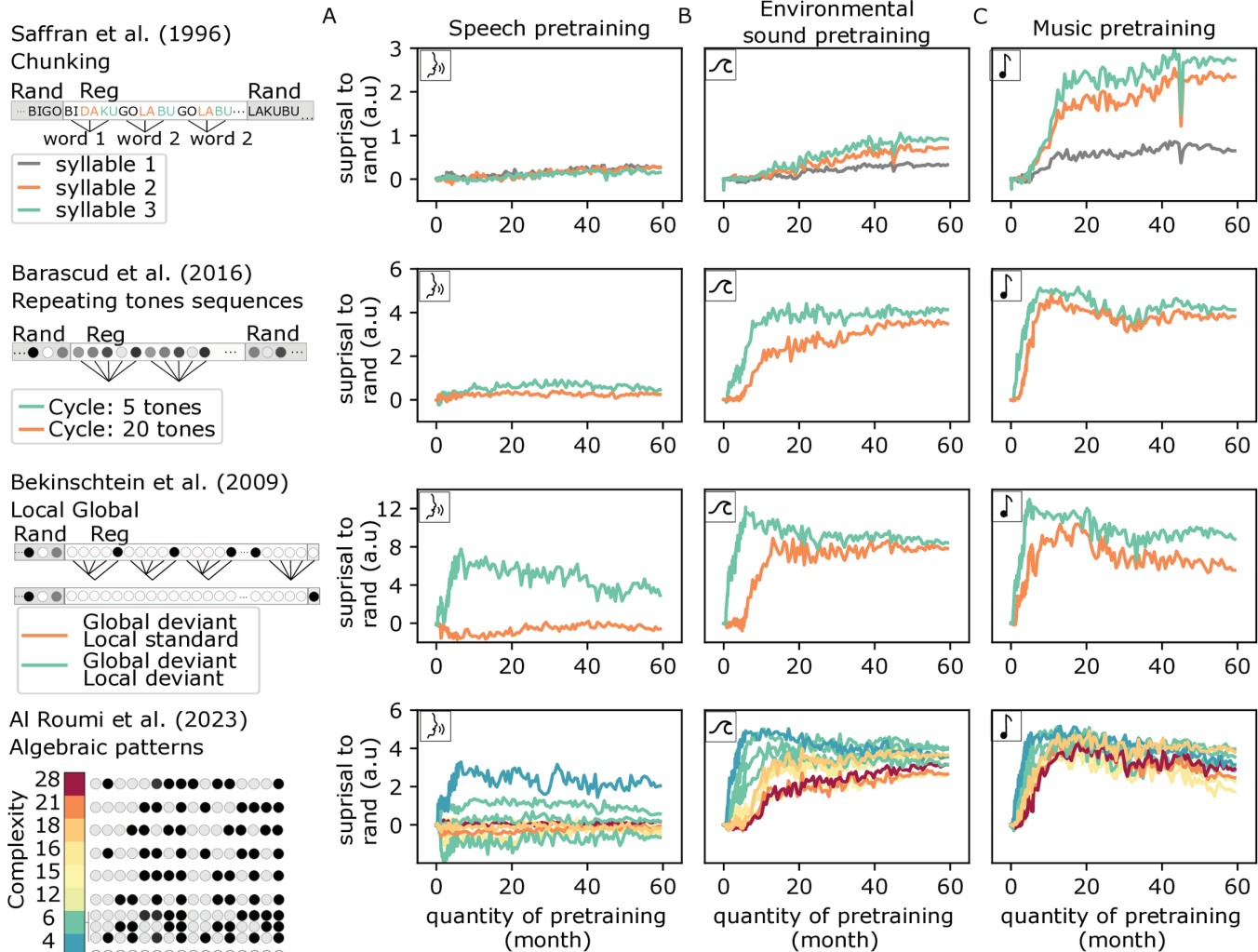

**Fig 3. Effect of the pretraining dataset on the emergence of structure detection.** We plot as a function of the amount of pretraining the ability of models trained on speech (A), environmental sounds (B) or music (C). Modeled studies: [8,14,21,23].

Next, we questioned if these results were robust to change in models' architecture. One architecture change that would not disturb model pretraining was to reduce the number of layers in the model. We trained tiny music and speech models with 3 instead of 12 transformer layers, our results were robust to these changes (Table B in S1 Material).

### Decoding the neural representations of auditory structures

Next, we wanted to get a better understanding of the sequence of operations performed by the model. To evaluate where auditory structures are represented in wav2vec 2.0, we trained a simple linear classifier on the model activations to detect whether the sound sequence was regular or not. We report the decoding curves in Fig 4. The results show that the violation to simple sequences can be linearly decoded from the first layers of wav2vec 2.0. By contrast, the violation of complex sequences can only be decoded in deeper layers. This result suggests that the model generates an increasingly abstract hierarchy of representations, which ultimately allows it to predict the probability of each sound segment. This result echoes neuroscientific findings, where the deviation to simple auditory regularities is computed before the deviation of complex auditory regularities [8,21].

## Discussion

Overall, our analyses demonstrate that structure detection can emerge from self-supervised learning: A generic learning objective—unmasking sounds—allows wav2vec2.0 to perform "in-context" detection of auditory structures, and this, without any additional learning or fine-tuning. Like humans, the spontaneous ability of a model to detect auditory structures depends on their algebraic complexity.

Together, these results challenge radically rationalist view, for which the neural representation of algebraic structures results from a built-in and specific mechanism. First, structure

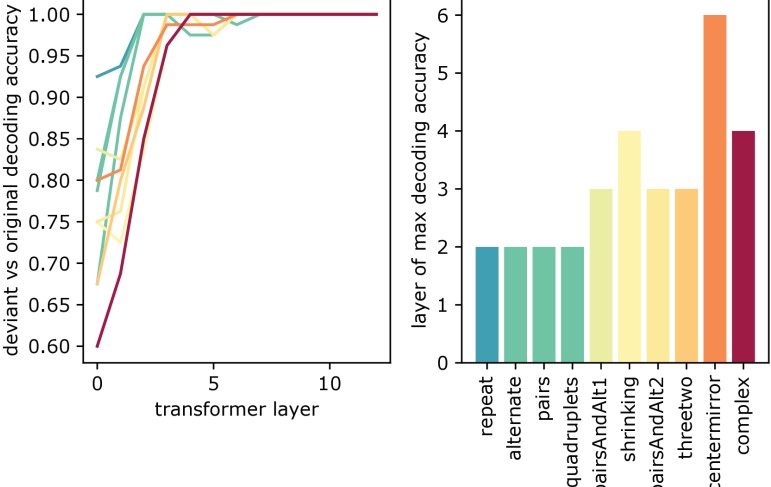

**Fig 4. Decoding regular vs non-regular for each model layer.** Left: decoding accuracy of deviant versus original tones. The classifier is learned over all types of sequences and evaluation is separated across different types of sequences. The dataset are composed of 1000 sequences, with the 500 test sequences being the exact complementary to the training sequences (inversion of the role of each tone). Deviant tones are detected in earlier layers for simpler sequences. Right: earliest layer at which the decoding accuracy is maximal, for each type of sequence.

detection can be discovered through pretraining and thus needs not be an axiomatic property of the network (Fig 2).

Second, and to our surprise, wav2vec2.0's ability to detect auditory structures does not appear to improve with speech exposure (Fig 3). While abstract structures are undoubtedly critical for syntax and language, these results suggest that the link between speech and the detection of auditory structures may be more tenuous than previously assumed (e.g. [4,6]). Third, both the surprise and the internal representations of these artificial neural networks correlate with the algebraic complexity of the auditory sequences. In particular, simple and complex auditory structures seem to be represented from early and deep layers, respectively. This result is consistent with previous neuroscientific findings. For example, Bekinschtein et al [21] showed that local auditory violations are detected early on (120–200 ms) in the auditory cortex, whereas global violations are detected later on (>300 ms) in the associative cortices. Overall, these elements thus show how modern self-supervised architecture may suffice to explain a wide range of behavioral and neuronal findings.

Two non-mutually exclusive interpretations may explain the unexpected dissociation between structure detection and language.

First, our models pretrained with speech sounds may learn distinct types of regularities compared to those pretrained with music. Unlike music and many natural sounds, speech does not have many repetitions [3]: entire sequences of tones can be repeated in musical pieces, whereas natural speech only rarely consists in repeating the same words and sentences. To verify this hypothesis, we compare the regularity of musical and speech sounds in the methods. This lack of repetition, or "asynchrony" of speech may thus lead the neural network to discard repetitions and focus on other statistical relationships, like meaning, syntax, and prosody. On the contrary, the "synchrony" of musical and of many natural sounds could favor the emergence of repetition detection, and, by proxy, the emergence of an efficient system of representations to detect and compress [8] these repetitions. A second, alternative, interpretation could relate to an inherent weakness of Wav2vec2: unlike text-based models, such speech-based model appears to lack many high-level syntactic and semantic properties (e.g. [39,52]). In contrast, the human brain may use additional algebraic representations to detect syntactic structures and recycle them to detect any auditory structure. This possibility, however, would need to reconcile the fact that mathematical, rather than language areas, seem to represent the algebraic auditory patterns [8,53].

In this work, we systematically varied the type and the amount of auditory exposure used for pretraining. Although unlikely, some effects could be due to the precise nature of the datasets used for pretraining. Additionally, several other components remain to be tested. First, all of the presently tested models used the same learning objective: unmasking. Yet, there exist other unsupervised objectives, such as causal predictions [54], compression [55], multi-losses and Hebbian learning. Second, we did not vary the architecture [41]. In addition, Wav2vec2 presents several implausible features. We report in supplementary materials, (Figs I–O in S1 Material) , several weaknesses of the model. We observe that at large (>200 ms) inter-stimuli intervals, the model computations are biased by normalization operations (Fig I in S1 Material). This normalization also leads the model to process differently sequences that differ minimally. These effects lead to surprising behavior, which can be resolved by the introduction of a few-shot protocol, introduced in the supplementary materials (Figs J–O in S1 Material). More problematically, the architecture uses an auditory buffer which gives it a perfect short-term memory over the presented sound waveform (e.g. 20 s of sounds). On the contrary, it lacks any long-term memory module to recollect past episodes. These features are at odds with humans [56]: For example, humans can partially memorize random 20-tones

sequences over several weeks [27,57–59], presumably thanks to a dialogue between the hippocampus and the cortex [60–63]. Overall, it is thus unclear whether our few-shot experiments do correspond to the memorization strategy effectively implemented in the brain. The fragility of the model behavior in the few-shot experiments, compared to the one-shot experiment, strengthens the idea that it misses modules for long term memorization.

Finally, we showed that self-supervised learning of naturalistic stimuli provides a sufficient basis for the detection of complex auditory structures. Yet, a core question remains: why then, are humans so much more efficient at such tasks than other animals? First, this cognitive advantage may not be that clear-cut.

While the seminal work of Saffran has triggered a wave of research demonstrating that "chunking" and statistical learning is not specific to humans, more recent work suggests that the detection of abstract structures may require a special set of computational operations (Dehaene et al. 2015 [4]). These landmark paradigms all illustrate the detection of different sets of abstract structures. The fact that they require special mechanisms remains speculative, but some authors have argued that humans show much faster and stronger abilities than animals on those tasks. [50,64] demonstrated that macaques encode abstract regularity in the local-global paradigm as well as numbers but, unlike humans, no conjunctive representations of those. [51] demonstrated similar encoding of local and global abstract regularities in the smaller marmoset brains. In the even smaller mice brain, [65] observed strong and generalized responses to local regularity but only weak and not generalized responses of a rare cell type to global regularity. In simpler tasks, interspecies differences also exist: ferrets do detect repeating sequences, but unlike humans [23], this ability is limited to seven tones [66]. These interspecies differences question the factors that govern the emergence of the ability to detect abstract regularities, which we explore throughout the article.

Overall, we can thus speculate that not only the amount of stimulus exposure, but also the sheer size of the cortex may be critical to zero-shot detect complex auditory structures. With the increasing size of large language models, some reaching 100 billion artificial units, it will be important to evaluate how network size and stimulus exposure synergize to spontaneously represent stimuli as abstract structures.

Large language models are notoriously less efficient than children when it comes to learning: while these algorithms require billions of words before being able to articulate coherent sentences, children learn to speak with a daily exposition of 13 thousand words [67]. Interestingly, the present study focuses on a different type of algorithm, namely a self-supervised audio model. Like children, this algorithm can be exposed to a lot of sound, which can be directly related to age, if we omit, for simplicity, moments of silence and sleep. Note that the comparison is less meaningful for music models, as these stimuli are comparatively rare in the day of a child.

One limitation of the present study, is that it primarily focuses on the *ability* of the model, but not on its internal mechanisms. Consequently, it remains unclear *how* abstract sequences are effectively represented in the neuronal activations. One interesting possibility is that these abstract structures are represented via a subset of neuronal activations that embed the underlying algebraic structure. Indeed, large language models have recently been shown to represent syntactic trees within a specific subspace where words are distant from one another as they would in their dependency trees [68]. As wav2vec 2.0 generates neural activations that are linearly aligned to those of the brain [39,69,70], the present approach offers a promising path to understand how the human brain represents abstract auditory structures.

Alternatively to these self-supervised emergences, previous works have measured how the training environment impacts supervised model processes. Notably, [71] revealed that human-like sound localization only emerged in environments including reverberation, noise,

or natural sounds. [72] stressed the importance of training the model in noisy speech environments to better predict brain responses to speech. The common denominator of these works is to illustrate the importance of approaching a realistic developmental environment to better model brain and behavioral responses. Closer to our work, [73] measured the existence of units responsive to music shuffling at sizes up to 400 ms in convolution neural networks trained to classify environmental sounds.

Overall, the present work thus acts as a proof of existence: the ability to detect abstract structures [4,7] can spontaneously emerge in a generic neural network trained solely from passive exposure to naturalistic sounds. Importantly, the systematic comparison across models reveals several factors of structure detection emergence. Notably, the type of sound matters. In particular, both music and environment—but not speech—models ultimately detect abstract structures. However, the speed at which this ability emerges varies considerably. This result fits with the popular notion that music stimuli may have been optimized for the acquisition of abstract structures [3,6]. While this hypothesis remains open for humans, the present work shows how modern deep learning models provide a testbed to study this possibility. Perhaps more importantly, the quick development of efficient models and learning rules pave the way to test, empirically, how built-in and acquired mechanisms may contribute to humans' unique ability to represent complex structures.

## Methods

### Models

We study the ability of a self-supervised neural network model: wav2vec2 to perform abstract structure detection. The model is composed of a series of processing steps organized in layers, which progressively transform the 1-dimensional auditory input at 16,000 Hz into a 768 dimensional vector sampled at 50 Hz. Each input is consequently transformed into a sequence of vectors, hereafter referred to as "latent". Self-supervised learning consists in building latents such that they can be predicted from their context. For wav2vec2, this objective is achieved with masking and contrastive learning. By evaluating the model at different training step, we can identify the amount of data required for the emergence of structure detection. In wav2vec2, the architecture is split into two major sets of layers. First, the sound is filtered and downsampled by a series of convolutions layers [74], typically known to efficiently learn temporally local patterns of sounds, like a pitch, or the frequency of a tone. Second, the resulting latent vector can be input to a transformer, typically known to efficiently combine contextual information to learn long-range dependencies, like the syntactic structure of a sentence.

### Models pre-training

Models are pretrained with Huggingface's distributed trainer [75], on a set of 64 V100 GPUs (8 gpus per node) for 100 000 steps of gradient descent, with a batch of size 4 in each GPU (each training took approximately 4 days). The training follows all recommendations and hyper-parameters used to train the base Wav2vec2 model in [41]. Notably, we make several modifications to the implementation of [75] since it missed some important elements of [41]. First, we scale the gradient of the encoding features by a factor of 10, which has for effect of slowing down the learning of the encoder compared to the transformer part of the architecture. Then, we multiply the mean of the squared encoding features (the features at the output of the convolution) by a factor of 10 and add this value to the total loss as a regularizer. If we didn't have these scalings, the model activity would either diverge or collapse during

training. We train with input sounds truncated to have a maximal size of 20s and a minimal size of 2s. For each dataset, we pretrain two models from different random initialization and random seed throughout pretraining. The results were all consistent across the pairs of models, demonstrating that the difference of results did not arise by chance. Moreover, the losses of the models during pretraining were similar across the 8 trained networks. To make sure our models had converged to reasonable solutions, we tested their classification performances on speech ( classification of logatomes), music (classification of musical genres), and environmental sounds (classification of environmental scenes). More precisely, for the speech test, we took a subset of the OLLO2 dataset [76]. The model is trained to predict a set of logatomes identity (9 distinct logatomes) from a subset of speakers and we test its ability to generalize across speaker. More precisely, we learn a cross-validated data normalization and logistic regression from the model time-averaged response to each sound and the class label. Scores are collected across cross-validation folds and across layers of the model and averaged. A similar approach is used for the music test with the GTZAN dataset [77], where the model needs to classify 30 distinct musical genres. We make sure to use error-free cross-validated folds [78]. Last, we use the ESC-dataset [79] for classification of environmental scenes. We here test the unsupervised model representations at the end of pre-training (no fine-tuning). Remarkably, models pretrained on a specific type of data performed better on the respective test than models pretrained on a distinct type of data (Fig H in S1 Material). Additionally, models trained on everything performed well on all the test. These observations confirm that models have converged to reasonable solutions and are able to perform the contextual processing required for these classifications.

All model weights of all checkpoints and the exact datasets used for pretraining are available at https://huggingface.co/NDEM.

## Pretraining datasets

We pretrain on the Librispeech [42], FMA [43] and Audioset datasets [44], as well as a mixture of these 3 datasets (Table 1). To generate the mixed dataset, we randomly picked 300 hours of sounds from each dataset.

The Audioset dataset contains low quality audio files, with sometimes loose annotations [73]. We removed from the dataset all audio files that were annotated with speech or music-related labels. The exact set of annotations we used is available as CSV files at https://huggingface.co/datasets/NDEM/dataset-audiosetfilter/tree/main/filters, the Hugging-Face repository of this dataset (speechlabels.csv and musiclabels.csv). To verify the quality of the dataset, we then manually listened to a subset of the training dataset, containing 150 files. Of those files, 9 (6%) of them included music, with 4 of these music tracks blended in the background and made of non-repetitive experimental samples. 12 files (8%) contained clear speech. In all the remaining files, a large amount (128 (45%)) contained nonmusical sounds but with clear temporal structures. Example of sounds with temporal structures included bird chirp, dog barking, tapping sounds, or repetitive mechanic sounds like motors. Other sounds could contained textured sounds, or a seemingly random sequence of unrelated sounds.

**Table 1. Comparison of audio datasets in terms of duration and number of files.**

|  | Librispeech | FMA | Audioset |
|---|---|---|---|
| **Number of hours** | 951 | 587.9 | 622.1 |
| **Number of audio files** | 277710 | 106395 | 227700 |

Based on this manual verification, we concluded that the contamination of the Audioset dataset with several musical events is neglectable compared to the other natural sounds.

## Sequences generation

Syllables were generated using the MRBOLA synthesizer, through the python package vox-populi. We used the English voice with a speed of 160. Each synthesized syllable was generated to be of duration 200 ms, re-sampled at 16000 Hz, and normalized to have a root-mean-square of one. We also removed any pitch modifier from each syllable during synthesis.

Tones were generated using librosa [80] and lasted 50 ms. Both tones and syllables were multiplied at their onset and offset with hanning windows (i.e. raised cosine window) of size 5 ms to avoid artefactual clipping. For Figs 2 and 3 we used sequences without any silences between tones or syllables. For, and only for, Fig I in S1 Material we removed the waveform normalization. With waveform normalization, the model is not robust to repeated silences and it became unable to detect structure at ICI larger than 0.2s. Without waveform normalization, the model is still not robust to silences but its performances decreased at a larger ISI.

## Loss measurements

Evaluating correctly the loss  of a masked network like Wav2vec2 requires some careful modifications to its original code base and careful experimental design. We have not seen earlier descriptions in the literature of these points of attention. To facilitate further research, we describe in detail the steps taken and provide our implementation of the needed modifications.

Wav2vec2 is equipped with a contrastive loss, as such it requires the selection of a set of masked latent vectors and for each of these latents a corresponding number of latents distractor from the same acoustic context, termed negatives. Each latent is sampled every 320 time-step i.e. every 20 ms (50hz), and gather information across a receptive field of 400 time-step.

To prevent leakage of information when computing Wav2vec2 loss, it is extremely important to mask all latents whose receptive field overlaps with the presence of a certain sound. When computing the loss for a certain sound event, we, therefore, masked all latents whose receptive field overlapped with this event. We computed a loss value for all these masked elements but kept and summed only the ones whose receptive field was fully contained in the sound event, producing a final loss value for this sound event.

Let us go through an example: given a 10-second waveform $w \in \mathbf{R}^{160000}$ discretized at 16000 Hz, the model computes a latent speech representation of dimension 512 $q \in \mathbf{R}^{(512,499)}$ at 50 Hz (the exact number of latent is 499 here, and can be computed from the strides and kernel sizes of the successive convolutional layers). These latents are separated by 20 ms, but they are the result of successive convolutions, such that their value is influenced by initial waveform values of up to 25 ms in the future. Consequently, to measure the surprise of a tone happening at 2s and lasting for 50 ms (2 to 2.05) it is not sufficient to mask the 100-th latent ( receptive field 2 to 2.025). We need to mask the latent 99 (receptive field 1.98 to 2.005), 101 (receptive field 2.02 to 2.045), and 102 (receptive field 2.04 to 2.065), but not the latent 98 (1.96 to 1.985) or 103 (2.06 to 2.085). Additionally, one needs to set to 0 all sound latent happening after 102, such that the relative positional embedding present in Wav2vec2 does not also introduce leaking of future information.

The contrastive loss contrasts the prediction over the masked latent with its true unmasked value and a set of other unmasked latent values, termed negatives. To obtain reasonable values of the loss, the set of negatives should be shared across masked latents, solely removing the

negative that matches the masked latent. Otherwise, the loss might purely reflect the acoustic distance between the embedding of the acoustic events composing the set of negatives. In that case, it would poorly reveal the network errors in predicting, correctly or not, the masked elements. To prevent such problems, our experimental design make sure to have sufficient negatives by using a random sequence composed of all the tones or syllables from the alphabet at the beginning of the sound. This design is therefore due to a limitation from the way negatives are sampled in Wav2vec2.

The model is evaluated in a partially causal way. More precisely, to measure the response to each tone, we set to 0 the activity of all future latent, critically before the positional embedding is computed. Note that it only makes the model partially causal because we do not: (1) set the attention masks used by the transformer to be causal; (2) adapt the positional encoding to take into account this causality. The reason is that Wav2vec2 is pretrained in an acausal regime, therefore it showed poor performance when evaluated with causal attention masks, which is the standard way to make a model causal. Nevertheless, the loss we measure remain only influenced by past sounds and is consequently a causal measure. Future work will adapt and pretrain novel models to solve these issues. These models will be able to perform in a single pass all causal predictions over a sound.

We evaluate the model over 371 checkpoints across the 100 000 training steps. More precisely we save the checkpoints from steps 1 to 100 by step of 1, from 100 to 1000 by step of 10, from 1000 to 10 000 by step of 100, and from 10 000 to 100 000 by step of 1000. This schedule allowed us to have a precise overview of the beginning of training where most of the interesting dynamics occur while leaving the possibility to observe long term dynamics.

We would like to note that there are two limitations concerning the probing of the model contrastive loss. First, the model has to be probed as many times as there are sound elements. Second, a decision to put a mask over certain tones is explicitly made, which acts as a prior that guides the model behavior. Future architectural work will consequently resolve these two issues. For the first, by developing a transformer model allowing parallel query of the loss for a set of sound elements partitioning a sound. For the second, by developing internal models that predict the mask to be used in a particular context. The first development is a matter of architectural tweak, slowed down by hardware bottlenecks. The second will require more research, for example with the use of oscillatory mechanisms to guide masking. This matter is pressing, as loss probing with masked auto-encoder is computationally very inefficient. The probing of the network activity and loss required around 30 000 hours of GPU compute time during the project, which could not have been done without a cluster. We estimate the carbon footprint of the project to be around 0.6 tones of $CO_2$, based on the grid consumption value provided in [81].

## Complexity metric

We use the complexity measure investigated in [82] and [8]. The sequence complexity was measured as the minimal length needed to write the sequence in a certain language, termed binary Language of Thoughts (LOT). We use the software from [82] to evaluate the LOT complexity of our novel sequences.

## Decoding regular vs non-regular

Next, we wanted to get a better understanding of the sequence of operations performed by the model. To evaluate where the model represents auditory structures, we trained a linear classifier to distinguish deviant versus non-deviant tones from the model activations. For that

purpose we generated for each sequence of Al Roumi et al. [8], 10 trials and their complementary. 20 tones were sampled on a log space from 222 to 2000 hz, the first 10 tones being set as low and the last 10 tones as high. For each train trial, the A tones were taken from 5 possible high tones and 5 other possible low tones. For each test trial, the A tones were taken from 5 other possible high tones and 5 other possible low tones. In both cases, a high A tone was paired with a low B tone, and a low A tone was paired with a high B tone. Finally, we added in the test trials each complementary trial from the train trials, and from the train trials each complementary trial of the test trials. This prevented overfitting of the decoder to tone frequencies.

## Music is more regular than speech

To compare the auditory regularity of musical pieces versus speech, we analyzed (1) the phonemic transcription of a 10h subset of the librispeech dataset [42], and (2) the MIDI files of LAKH musical dataset [83]. For each sequence of phonemes or notes, we then computed the normalized LZ-complexity [84,85], i.e. a repetition metric that quantifies the number of different substrings encountered as the sequence is iterated from the start to the end, corrected for the sequence length. We cut musical pieces so that they all had the same number of elements as the mean length of phoneme sequences The results show that music sequences have a lower LZ-complexity (m = 0.21, std = 0.05) than phonetic sequences (m = 0.75, std = 0.05). This result confirms that musical sounds are more regular than speech sounds in these corpora.

## Supporting information

**S1 Material.**
(PDF)

## Acknowledgments

The author would like to thank Stéphane Deny, Fosca Al Roumi, Christophe Pallier, Linnea Evanson, Théo Desbordes, Sam Norman-Haignere, and Stanislas Dehaene for comments on the work and manuscripts. This work was granted access to the HPC resources of IDRIS under the allocations 2023-AD011014524 and 2022-AD011013176R1 made by GENCI (P.Orhan). The author would like to thank IDRIS for their support on the Jean-Zay cluster.

## Author contributions

**Conceptualization:** Pierre Orhan, Yves Boubenec, Jean-Rémi King.

**Data curation:** Pierre Orhan.

**Formal analysis:** Pierre Orhan.

**Funding acquisition:** Pierre Orhan, Yves Boubenec, Jean-Rémi King.

**Investigation:** Pierre Orhan.

**Methodology:** Pierre Orhan, Yves Boubenec, Jean-Rémi King.

**Project administration:** Yves Boubenec, Jean-Rémi King.

**Resources:** Pierre Orhan.

**Software:** Pierre Orhan.

**Supervision:** Yves Boubenec, Jean-Rémi King.

**Validation:** Pierre Orhan, Yves Boubenec.

**Visualization:** Pierre Orhan.

**Writing – original draft:** Pierre Orhan, Yves Boubenec, Jean-Rémi King.

**Writing – review & editing:** Pierre Orhan, Yves Boubenec, Jean-Rémi King.

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
