## [Decision Letter · Decision Letter 0]

11 Mar 2025

PCOMPBIOL-D-25-00137

The detection of algebraic auditory structures emerges with self-supervised learning

PLOS Computational Biology

Dear Dr. orhan,

Thank you for submitting your manuscript to PLOS Computational Biology. After careful consideration, we feel that it has merit but does not fully meet PLOS Computational Biology's publication criteria as it currently stands. Therefore, we invite you to submit a revised version of the manuscript that addresses the points raised during the review process.

Please submit your revised manuscript within 60 days May 11 2025 11:59PM. If you will need more time than this to complete your revisions, please reply to this message or contact the journal office at ploscompbiol@plos.org. Please include the following items when submitting your revised manuscript:

We look forward to receiving your revised manuscript.

Kind regards,

Yuanning Li

Academic Editor

PLOS Computational Biology

Marieke van Vugt

Section Editor

PLOS Computational Biology

**Additional Editor Comments:**

The reviewers were in general positive about the manuscript, but raised concerns about the methods and analyses. Addressing these concerns would rule out potential confounding factors and strengthen the major claims.

**Journal Requirements:**

5) Please ensure that the funders and grant numbers match between the Financial Disclosure field and the Funding Information tab in your submission form. Note that the funders must be provided in the same order in both places as well.

**Reviewers' comments:**

Reviewer's Responses to Questions

**Comments to the Authors:**

Reviewer #1: In this work the authors set out to study whether representations of algebraic auditory structures can emerge in a model trained in a purely self-supervised fashion without enforcing specific rules and constraints to make it possible. In the bigger picture, the answer to this question aims to address the bigger debate between "radicalist" and "empiricist" views on how humans learn to detect abstract structures in sound and other sensory modalities. The authors take a novel approach to the problem by using zero-shot contrastive loss of individually masked units of sound to measure model surprisal to stimuli, and then using that surprisal to emulate psychoacoustic experimental paradigms to measure the abstraction of auditory structures as studied in humans. The results of the study make a strong argument for the empiricist view.

Overall, I found the paper highly engaging in terms of novelty of approach and results, and rigorous in methodology. I did not come across any major issues, but I do have a few minor comments that I hope will serve to improve the manuscript even further.

Minor comments:

1. One of the most interesting parts of the results for me was the finding that models trained on speech alone did not learn to extract abstract structures. While the authors have dedicated a section to this in the results and also address it in the Discussion and supplemental materials, I think there is space for more exploration here. There are a few examples below. I do not expect them to be addressed with new experiments, but I think some more general discussion on this point is in order.

a. The authors name one possible reason for the finding which is that the speech dataset used for training is not as "repetitive" as the music dataset, which they also show quantitatively. Does this suggest if a more repetitive speech dataset was used for training, like a larger version of the TIMIT dataset, it would show a higher degree of abstraction? What about songs?

b. The authors show that all four models trained perform reasonably well on some high-level classification tasks involving speech, natural sounds, and music (Suppl Fig 8), but these are perhaps too high-level. For example an automatic speech recognition model will learn to ignore most sources of non-speech sounds as they are noise with respect to the task of speech recognition. Similarly, the Wav2Vec2 trained on speech might be suppressing/normalizing the non-speech sounds in the stimulus, which would include most of the experiment data which consist of tones. One way to test that this is not the case is to measure the mask reconstruction accuracy of the speech model on a few very simple patterns of alternating tones (ABABAB...) since that is the stimulus for 3/4 tests. Something like this would show if the speech model is capable of sufficiently representing the building blocks of the test stimuli.

c. Connected to point (b) above, the tests are made up of sequences of tones which are mostly found in music. Could this be the reason why musical exposure shows the biggest promise? What if the stimulus units in all cases were syllables or pieces of stationary environmental sounds?

d. The results for the models trained individually on the three types of data are compared, but I would also suggest comparing results between the model trained on music only vs the one trained on a balanced dataset.

2. On lines 201-202 of the manuscript, I did not follow how the authors came to the conclusion that "3 months is therefore an upper limit on the amount of data needed for this detection". Did this follow from the finding that after 3 (audio-)months of pre-training the model it was able to detect global deviant - local standard? Does this not depend on data variety, loss function, and model architecture/size? Or did it follow from the study that showed children at 3 months old are able to learn such patterns? Either way, I think the sentence could use further clarification.

3. There is a surprising result in Figure 4 that is worth mentioning. And that is the case of the simplest algebraic pattern – repeat. To my understanding the decoder here is a binary classifier, which makes the random baseline accuracy 50%, correct? So when the decoding accuracy is ~5% for the first transformer layer, that means if you flip the prediction it would be correct 95% of the time. Is this a bug, is the dataset that this decoding is being tested on unbalanced, or something else? Because as is, it looks like the simplest algebraic pattern emerges deepest in the model, if we define emergence as crossing threshold of X% accuracy where X < 90.

4. While the scientific content of the article was quite engaging to read, I think the language could have been improved in a few places. Specifically cases of grammar mistakes, typos, and sentences that were too long. These made the paper difficult to follow at times.

Reviewer #2: This paper investigates whether Wav2Vec2, a self-supervised deep learning model, can detect complex auditory structures similar to human cognitive abilities when trained on natural speech, music, or environmental sounds. The findings demonstrate that the models progressively acquired the ability to detect increasingly complex auditory structures with continued training. Notably, this ability did not emerge when the model was trained solely on natural speech and developed more rapidly with music pretraining compared to environmental sounds. The paper situates itself within the "rationalists vs. empiricists" debate and appeals to a broad audience in cognitive science. I'm generally positive for endorsing this paper for publication, but the following comments should be addressed before publication.

Major:

1. The syllable stimuli used for testing word segmentation were generated with a synthesizer, with pitch removed to mirror the original setting of Saffran et al. (1996). However, subsequent work by (Yang, 2004) has shown that stress patterns play a more significant role than transitional probabilities between syllables in real word segmentation. This could explain the model's failure to perform chunking when trained solely on speech data. It would be interesting to see how the model performs with three-syllable sequences featuring primary stress on the first syllable.

2. The finding that music pretraining led to better structure detection compared to speech and environmental sounds is intriguing but lacks a sufficient explanation. Is there any metric available to quantify the similarity between the training data and the test stimuli? Could it be that the music dataset is, in fact, more similar to the synthesized test stimuli?

3. I understand that it is interesting to demonstrate how models trained on natural sounds can generalize to artificial structure detection. However, would it also be worthwhile to include results on the model's performance in detecting natural structures, such as syntactic constructions?

Minor:

1. The labels on all line plots in Figures 2 and 3 indicate "surprise" or "surprisal," while the captions describe the metric as the "difference between the mean contrastive loss." This seems to deviate from the canonical definition of "surprisal," typically defined as the negative log of a probability distribution. Additionally, the Methods section mentions the LOT complexity metric. Could you clarify where this metric was applied in the analyses?

2. The authors deliberately exclude LLMs for their use of "unreasonably large and symbolic datasets." However, it might be valuable to include the performance of these LLMs on the testing data for comparison.

Yang, C. D. (2004). Universal Grammar, statistics or both? Trends in Cognitive Sciences, 8(10), 451–456.

Signed: Jixing Li

Reviewer #3: The paper is an interesting set of computational experiments studying how differences in self-supervised training of DNNs can lead to different emergent properties in detecting structure of the input. The authors relate these properties to well-studied phenomena found in humans that emerge over development. Overall, I found the paper interesting, but there are a few major concerns regarding possible confounds of the experiments that I think should be addressed prior to publication.

Major Concerns:

1) I am concerned that the reported results with environmental sound training is contaminated by the presence of music in the training dataset. The authors state that their model is trained on “environmental sounds” by excluding music and speech sounds from AudioSet using the taxonomy, however the AudioSet labels are sloppy and if a label is not present on a clip that does not guarantee that the category is *not* present in the clip (quality ratings for the labels are for positive labels). In fact, there are a large number of AudioSet clips containing music but not labeled as so. This has been pointed out for other similar studies, for instance: https://www.nature.com/articles/s41467-023-44516-0 where they report that in the *balanced* dataset about 4.5% of the clips have music (and I believe the listed clips are on their GitHub). I suspect this is even higher for the unbalanced dataset. Did the authors ensure that their training set has NO music? If not, to make the claim about the differences in training environments, I believe a new model must be trained on data known to contain no music (yes, I acknowledge this might be quite time-consuming, sorry). It unfortunately could be the case that it has some music but less which is driving the similarity to the musical training. Further, I could not find the list of categories that were excluded from AudioSet for training. This *must* be reported for reproducibility (for instance, were all of the musical instruments excluded? What about things like telephone ringtones?)

2) I am concerned about differences across the 4 tests of algebraic structure being driven by differences in acoustic cues. Short tones vs. syllables could be represented very differently by the neural networks (for instance, in a speech-trained network, tones are out of distribution, and so all tones may have a high loss). To evaluate which parts of the results are driven by low-level cues vs. the higher-order algebraic structure, perhaps the authors could construct two control experiments (1) for chunking that uses tones rather than syllables and (2) for at least one of the other experiments that substitutes syllables in for tones. This would show which differences in Figures 3 and 4 (and similar SI Figs) are due to the underlying audio rather than the actual structure the authors are trying to test.

3) Claims about training dynamics may be dependent on the choice of learning rates. The learning rate schedule and chosen optimization method for a given model will significantly change the trajectory of training, and different training datasets or models can require different learning rates for end convergence of good representations. How were the learning rates determined for each experiment? Do the results depend on these choices? To make strong claims about differences in training datasets (ie Fig 3) it seems that one would need to show that the trends emerge regardless of optimizer. One way to extend this may be to train the models with a different architecture, as generalization across architectures makes an even stronger claim that this is about the training dataset and not decisions during model pre-training.

Minor Comments:

- Methods: I could not find the exact amount of unique data included for each of the sound sets (I guess this is ~300 hours of each set, according to line 101, but I found this description confusing). The axes are related to the hours of pretraining, but it is important to know how many passes through unique examples are included in each experiment. It might be helpful to include this in some way for the main text figures.

- General Wording: The abstract and introduction contain some jargon that is not well defined and may not be common to some more general readers. For instance “merge” and “neural recursion” are a bit distracting at the start of the paper without giving the readers a little more context.

- Citations: There is quite a bit of work in the Computational Cognitive Neuroscience world that touches on how differences in training environments shape the behavior and internal representations of neural networks. Although much of this work focuses on supervised networks, the general claims about training environments seem relevant to this work. Some examples here, but I encourage the authors to do a further literature review:

https://www.nature.com/articles/s41562-021-01244-z

https://journals.plos.org/plosbiology/article?id=10.1371/journal.pbio.3002366

https://www.nature.com/articles/s41467-024-53147-y

https://www.nature.com/articles/s41467-023-44516-0

Typos:

-Line 57: “critical to build system” -> “critical to build systems”

-Line 162: ‘ is facing the wrong way for Repeating tones

-Line 382: Some words are missing or out of order in the first part of this sentence.

**Have the authors made all data and (if applicable) computational code underlying the findings in their manuscript fully available?**

Reviewer #1: Yes

Reviewer #2: **No: **p.16, 370: "All model weights of all checkpoints and the exact datasets used for pretraining are available upon reasonable requests to the author".

Reviewer #3: Yes

PLOS authors have the option to publish the peer review history of their article (what does this mean?). If published, this will include your full peer review and any attached files.

Reviewer #1: No

Reviewer #2: **Yes: **Jixing Li

Reviewer #3: No

**Figure resubmission:**
---

## [Decision Letter · Decision Letter 1]

26 Jun 2025

Dear Mr orhan,

We are pleased to inform you that your manuscript 'The detection of algebraic auditory structures emerges with self-supervised learning' has been provisionally accepted for publication in PLOS Computational Biology.

Before your manuscript can be formally accepted, you will have the opportunity to revise according to the minor comments remaining from Reviewer 3. You will need to complete some formatting changes, which you will receive in a follow up email. A member of our team will be in touch with a set of requests.

Best regards,

Yuanning Li

Academic Editor

PLOS Computational Biology

Marieke van Vugt

Section Editor

PLOS Computational Biology

I suggest the authors consider adding to the limitations regarding Reviewer 3's comments.

Reviewer's Responses to Questions

**Comments to the Authors:**

Reviewer #1: I appreciate the authors' thorough responses to my comments, including running new experiments. I believe the added and modified experiments and the other modifications to the manuscript have reasonably answered the major concerns that were raised and generally improved the robustness and clarity of the paper. Therefore my recommendation is to accept this work for publication.

Reviewer #2: I appreciate the authors’ thorough efforts in addressing my earlier comments and have no further concerns.

Reviewer #3: Thank you to the authors for addressing most of the concerns in the paper. I think that the submission is significantly better after the changes and additional experiments. The 3x3x4 design seems like a good way to test the different aspects of the models.

One remaining consideration:

I appreciate the authors reporting the screening criteria for the sounds, but 6% of the sounds including music and 9% including speech seems non-negligible. While I agree in principle that it seems unlikely this contamination is driving the main results in the paper, I do think that the authors need to mention the data contamination as a limitation in the discussion section. For instance, adding one or two lines to the paragraph starting at line 291 stating that although it is unlikely, there could be some effects due to the precise nature of the datasets used for training (including data contamination). One future piece of work could train some of the tiny models while parametrically varying the amount of various types of sounds in the training dataset.

Thanks again for addressing the concerns and for sharing this interesting piece of work!

**Have the authors made all data and (if applicable) computational code underlying the findings in their manuscript fully available?**

Reviewer #1: Yes

Reviewer #2: Yes

Reviewer #3: None

PLOS authors have the option to publish the peer review history of their article (what does this mean?). If published, this will include your full peer review and any attached files.

Reviewer #1: **Yes: **Menoua Keshishian

Reviewer #2: **Yes: **Jixing Li

Reviewer #3: No

---

## [Editor Report · Acceptance letter]

PCOMPBIOL-D-25-00137R1

The detection of algebraic auditory structures emerges with self-supervised learning

Dear Dr Orhan,

I am pleased to inform you that your manuscript has been formally accepted for publication in PLOS Computational Biology. Your manuscript is now with our production department and you will be notified of the publication date in due course.

With kind regards,

Judit Kozma
